# Tensile Behavior of Joints of Strip Ends Made of Polymeric Materials

**DOI:** 10.3390/polym14224990

**Published:** 2022-11-18

**Authors:** Andrei-Marius Mihalache, Vasile Ermolai, Alexandru Sover, Gheorghe Nagîț, Marius-Andrei Boca, Laurențiu Slătineanu, Adelina Hrițuc, Oana Dodun, Marius-Ionuț Rîpanu

**Affiliations:** 1Department of Machine Manufacturing Technology, “Gheorghe Asachi” Technical University of Iași, 700050 Iași, Romania; 2Faculty of Technology, Ansbach University of Applied Science, Residenzstraße 8, 91522 Ansbach, Germany

**Keywords:** strip end, omega contour, dovetail contour, polymer, tensile, force, displacement, modeling

## Abstract

The strength of a joint between the ends of one or more strips can be improved by making the contours of the joint into the shape of either the Greek letter omega or of a dovetail. From the point of view of industrial practice, it is of interest to study the behavior of these joints under stretching demands. The emergence and development of additive manufacturing processes for parts made of polymeric materials has led to the idea of conducting experimental tests to highlight the behavior of omega and dovetail-type joints during the tensile test. For the tensile testing of some test samples in which omega and dovetail joints were used, a Taguchi array of type L18 was employed, with eight independent variables, one variable with a two-level variation, and seven variables with variations on three levels. As independent variables, the type of joint, the couple of polymer materials used to make the two components of the joint, some characteristic dimensions of the joint contours, and some input factors in the 3D printing process were established. The values of average force and average displacement at the peak were considered output parameters. The experimental results were mathematically processed, determining empirical mathematical models of the second-degree polynomial type. These models highlight the influence exerted by the considered input factors on the values of the output parameters.

## 1. Introduction

Belt-type strips can be used as transmission belts and in other products’ assembly or packaging activities. In this paper, a strip is considered a part where the length is much greater than the width, and the thickness is relatively low compared to the width. The strips are slightly rigid in the case of bending stresses exerted in a perpendicular direction to the surface, in the dimensions corresponding to the length and width of the strip. Most strip parts are subject to tensile stresses.

The development of processes for manufacturing parts made of polymeric materials through 3D printing has facilitated the development of research on how to join strip ends, whether made of the same or of different materials. The strips can be made of metallic or non-metallic materials, the last group being, for example, polymer materials. To increase the strength of the joint between strip ends, it has been possible to introduce an interweaving of several layers belonging to the two strip ends to be joined.

The strength characteristics of the joint of strip ends made by 3D printing depends on the joint’s shape and dimensions and the values of some input factors in the 3D printing process.

When using distinct polymer materials for the two strips, additional problems arise as to how they are joined. To obtain a strong joint when using the interweaving solution, a degree of chemical compatibility and a certain bonding of the polymer materials used in the 3D printing process is required. The use of interweaving must be applied in such a way as to avoid, as far as possible, too pronounced porosity or the appearance of cracks.

Problems specific to the behavior of strip end joints or other types of mechanical joints have been addressed by some researchers concerned with obtaining joints with higher mechanical strength.

Thus, Tiwari et al. analyzed the possibilities of making a joint starting from components manufactured by 3D printing from similar/dissimilar thermoplastic parts and using friction stir welding [1]. They focused on the optimization of the behavior of the joint from the point of view of weld strength, elongation, hardness, and desired flatness.

Lopes et al. studied the tensile strength of multi-material test samples made by additive manufacturing from polymeric materials [2]. The obtained results highlighted the importance of the influence of the interface formed upon the contact of some components made of different materials on the value of Young’s modulus and the value of the tensile strength of the multi-material test samples.

Chen and Zhang used 70/30 and 30/70 ABS/PETG blends to evaluate notched impact strength [3]. They observed that ABS/PETG (30/70) blends showed an obvious increase in impact strength. The addition of a compatibilizer can result in improved flexural strength.

The problem of the influence exerted by slicing parameters on multi-material adhesion mechanisms in the case of parts manufactured by 3D printing was addressed by [4]. They found that interlocking strategies can compensate for the disadvantages of less favorable thermodynamic and diffusion mechanisms.

Ribeiro and Sousa Carneiro considered the problem of interface geometry in the case of multi-material test samples produced by fused filament fabrication from polymer materials [5]. They used T-shape, U-shape, and dovetail shape joints.

Ando et al. have achieved a stronger horizontal adhesion and a weaker vertical adhesion in the case of the tensile test of some samples made by additive manufacturing of multi-colored polylactic acid materials [6].

Kluczyński et al. [7] used both non-destructive testing methods (laser amplified ultrasonography, microscopic observations) and tensile tests to study the behavior of joints generated by multi-material additive manufacturing. They found that an overlap connection of two parts made of polylactic acid can increase tensile strength by up to 14% and elongation by up to 22%.

Hasanov et al. addressed the problem of additive manufacturing of multi-material parts [8]. They appreciated that to obtain a joint with high mechanical resistance, it is necessary to consider the factors that influence the result of combining the two materials To lessen the negative effect of shrinking adhesion between the components of a joint made of chemically incompatible materials, Kuipers et al. proposed using a so-called interlaced topologically interlocking lattice [9]. They found that in this way, joints with a behavior close to that of dovetail in-interlocking could be obtained.

In this paper, the research results sought to highlight the influence exerted by the shape and dimensions of joints, along with the values of some input factors in the 3D printing process, on the tensile strength of samples made of pairs of distinct polymers. Joints in the form of the Greek letter omega and dovetail joints were considered.

## 2. Materials and Methods

### 2.1. Initial Considerations

When the problem of joining the ends of a strip or several strips arises, it is necessary to choose from several possible alternatives approaches. The simplest way to join is along a straight line perpendicular to the parallel edges of the strip. A slightly more complicated alternative, which increases joint strength by increasing the length of the joint area, could be along an inclined line to the parallel edges of the strip. In both cases highlighted previously, it is necessary to glue or weld those surfaces of the strip or strips to make the joint.

An increase in the tensile strength of the joint becomes possible by making a shaped area on one end correspond to the the same shape in reflection on the other strip end to which the first end is to be joined, as in a jigsaw puzzle. As such, several usable solutions have appeared. Two involve a joint along a contour similar to the Greek letter omega (Ω) or, respectively, along a so-called dovetail contour (Figure 1). Also, in this case, to increase the operational reliability of the joint, it is beneficial to use gluing or welding of the strip ends.

For both omega and dovetail-type joints, the mechanical stressing and breaking of the strip ends are affected by the dimensions that characterize these joints.

Thus, if we consider in both cases the presence of a “neck”, when the width *n_w_* of the neck is large to the width of the strip, it is expected that the lateral areas of the opposite end will bear an elastic deformation until the free exit of the profile in the form of an arc of a circle or an isosceles trapezoid of the lower end of the strip (Figure 2a); or even to the breaking of the two lateral areas of the upper end (Figure 2b). If the width *n_w_* of the neck is small compared to the width of the strip, it is expected that the strip end break will occur right next to the bottom end neck (Figure 2c).

The emergence and development of additive manufacturing processes have increased the number of possible alternatives for joining strip ends. It was possible to interweave some layers of the materials of the two ends of the strip (Figure 3). This situation will lead, on the one hand, to reducing the risk of one end of the strip moving to the other along a direction perpendicular to the strip. On the other hand, increasing the contact surfaces between the two ends of the strip could increase the joint’s mechanical strength.

An interesting problem is also the behavior of end joints of strips made of different materials when the joints are made by 3D printing. Some aspects of the behavior of such a joint will be considered next.

Multi-material models result from joining two or more bodies of the same part (i.e., meshes). Depending on the model, the joint surfaces have a simple flat or circular arc profile. Since each part has a material associated with it, the printer will only deposit the materials within the mesh boundaries. Depending on the part design, the materials can be deposited side by side, layered, or both [2].

To increase the strength of the bond between materials, slicing tools such as Cura provide a parameter called Merged Meshes Overlap (MMO), which allows materials to cross into each other to interweave, improving joint strength. For this study, the parameter was set to zero to control it via CAD modeling.

### 2.2. Materials

Three distinct materials were chosen: a blue acrylo-butadiene-styrene from REC, a black polylactic acid PA645 (i.e., a PA6.6-based material) from Taulman3D, and a red Polyethylene terephthalate glycol PETG. These materials were chosen based on their low compatibility and similar melting range. A literature review shows that these pure polymer combinations, ABS-PA645 [8], ABS-PETG [1], and PETG-PA645 [10], have poor bonding. However, research has shown that blends of ABS and PETG have better properties than the base materials [11].

### 2.3. Experimental Conditions

As is known, when the determination of an empirical mathematical model is intended to highlight the influence of different input factors in the investigated process on some process output parameters, the use of the least squares method may make it necessary to perform a large number of experimental tests to reach an acceptable confidence level for the identified empirical mathematical model.

However, through the so-called design of experiments, it has become possible to significantly reduce the number of experimental trials required so that an empirical mathematical model can be determined through the mathematical processing of the results [12,13,14]. Ronald Fisher proposed the first principles of using the design of experiment method in 1924. In 1935, Fisher even published a book entitled “Design of experiments”.

Starting in the 1950s, Genichi Taguchi identified a series of situations in which the number of experimental tests can be reduced yet further, without significantly affecting the quality of the identified empirical mathematical model [15,16,17].

An experimental research solution using a relatively small number of experimental trials corresponds, for example, to a Taguchi L18 array with one input factor at two variation levels and seven input factors at three variation levels [18,19,20]. Such a Taguchi L18 array will be used to determine an empirical mathematical model capable of highlighting the influence of some input factors in the 3D printing process on the average force at peak and, respectively, on average displacement at peak.

For the design of the experiment, a Taguchi L18 array with one input factor at two levels and seven input factors at three levels was considered.

The input factors are joint shape (*j*), material pairs (*m*), neck width (*n_w_*), neck height (*n_h_*), shape height (*s*), number of walls (*no_w_*), line width (*l*), and the overlap level (*o*) of the layers of the two parts being joined.

The values corresponding to the various levels of the input factors can be seen in Table 1. The first variable provides information on the joints’ shape, omega, and dovetail. The joint dimensions are defined by the neck width (*n_w_*), neck height (*n_h_*), and shape height (*s*). As seen in Figure 4, the contact interfaces are constrained symmetrically relative to the mid-plane of the test sample.

The last three input factors consider some characteristics of the 3D printing process of test samples. The number of walls (otherwise known as contours, perimeters, or shells) *no_w_* refers to the number of closed paths deposited along the edge of the part [21]. The line width or the extrusion width (l) is the thickness of an extruded filament that influences the number of passes required to fill a full-density layer. Good results are obtained with line width between 60% and 150% of the nozzle orifice diameter. Higher values of this ratio can be used, but the results depend on the extrusion subsystem and filament diameter. A wider line width decreases the manufacturing time [22] and increases the tensile strength [23]. However, wider width lines increase the heights of surface asperities generated, due either to larger gaps between filling lines and walls [24] or to under extrusion if the extruder cannot melt the filament and provide a constant material flow.

Tensile tests were performed using an Instron 4411 uniaxial testing machine with a load capacity of 5 kN. All samples were printed using a 3D printer, Ultimaker 3, in an enclosure after reaching 35 °C. Each run covered in the experimental array was printed in four or five replicates. The results of the average tensile strength tests were processed using the statistical tool Minitab 21.1. As a benchmark for the resulting samples, six groups of samples were printed having the following material pairs: ABS-ABS, PA645-PA645, and PETG-PETG, to compare both forms of interconnection. The other parameters were kept at the mid-level values (see Table 2, except for overlap, which was set to zero).

The conditions under which the experimental tests were carried out and the results of these experimental tests are included in Table 1. It is necessary to specify that the values of the output factors in the last two columns of Table 1 are average values for four experimental tests carried out with the same values of input factors. An image regarding the dispersion of the experimental results from the case of experiments no. 2 and 12 can be seen in Figure 5.

The overlap *o* (Figure 3 and Figure 6) between the female and male bodies was adjusted in the sample design stage. Thus, the overlap value of the merged meshes in Cura was set to zero. Alternate mesh removal is another parameter enabled when printing samples. When the meshes are overlapped, the slicer removes a part of the mesh to make space for the other sample’s mesh. In this way, the meshes alternate layer by layer, woven like the structure of the materials. An example of the effect of using the ‘meshes overlap with alternate mesh removal’ function enabled is shown in Figure 6.

As shown in the preview of the g-code, in addition to the interlocking geometry, the materials are also held together by the woven structure created by overlap and alternate mesh removal. It was hypothesized that the frictional force (generated by the stacked layers) would improve the tensile strength of the test pieces.

In this regard, the line width *l* could influence the value of the pulling friction force. As previously mentioned, if the line width *l* is larger, larger voids may occur—either when the lines fuse or due to the possible under-extrusion. This means the second material deposited in the overlapping region will fill the voids formed, creating a mechanical interlock between the layers. The values of the other input factors considered for printing were kept at a constant value and can be seen in Table 2.

The appearance of some damaged test samples as a result of the tensile test is highlighted in Figure 7. In the case of some test samples made from the ABS-PA645 material couple (experiments R1 and R10), the joint damage develops through the elastic deformation of some areas of the two components of the joint. This fact could be justified by the lower mechanical resistance of the materials of the two components. In the case of some test samples made of the ABS-PETG material couple (experiments R4 and R13), damage to the test samples was observed near the interweaving area over the entire width of the test sample.

### 2.4. The Use of the Finite Element Method to Model Some Aspects of the Tensile Behavior of a Joint

Finite element method (FEM) analyses were first considered to highlight stress and strain distribution on selected surfaces of the assemblies. From the 18 samples, two different experiments in terms of shape were selected: experiment R2, corresponding to an omega joint; and experiment R12, corresponding to a dovetail joint. These experiments led to severe damage to the test samples at the end of the tensile tests.

Some material properties were considered for ABS and generic ABS plastic PA645 and PET plastic. Such properties have been identified in Ansys’s Granta. We considered modeling new materials in Ansys’s Material Designer module derived from PET plastic with added uniaxial tension test data. That allowed us to design an anisotropic response based on the matrix of the material as mentioned above. The model was set to create a randomized unidirectional composite with default values for volume fraction, seed, and fiber diameter. It used conformal meshing, model set as anisotropic when computing with linear elasticity. Thus, we obtained the stiffness matrix. The results were exported to engineering data where we added the uniaxial tension test data in tabular form, given by the Instron 4411 equipment used for experimental tensile tests. Because both samples cracked, the model aimed to use a method of crack growth assessment to find out if the results matched in terms of direction and shape. Static Crack Growth was used for the possible estimates of the crack growth in case of fracture [25]. In the case of the female component, a quadratic mesh size corresponding to the mesh element of 1 mm was combined with a patch-conforming method based on the use of tetrahedrons, which together took into account the computational limits proposed. The physics preference was set to Nonlinear Mechanical.

The simulations took into account different fixed supports and forces that would be involved in the tensile tests. Since there were different layers, time substeps were used.

The best-case scenario involved different loads and supports for each layer. However, the results are intermediate and should not be used in new research without further refinement due to the designed material’s behavior which can be different in terms of fiber reinforcement and additives used by the manufacturer. Only the female part of the two distinct shapes were considered and put to the test by treating each strip end individually. Thus, we imposed supports and loads on each strip end for the omega and dovetail-shaped samples which received pre-modeled V-shaped cracks. Within Ansys Static Structural, we used the Fracture tool with the SMART Crack Growth condition based on Static Crack Growth with Failure Criteria Option set to Stress Intensity Factor with a critical rate of around 100 MPa. All cracks received their Coordinate Systems that were used inside the Pre-Meshed Crack tool with six solution contours. Fracture Controls in Analyses Settings were on, and Stress Intensity Factor enabled. The strip ends that were not cracked received force loads acting almost normally in the Y direction of the top face of the predesigned crack as the X axis points to the initial crack growth direction. This resulted in an asymmetric response using the Nonlinear Adaptive Region tool with the criterion set to Mesh Quality with the option for Skewness and the Jacobian ratio at default values.

The R2 sample exhibits severe failure, as shown in Figure 8. FEM yields a value of around 0.07 mm/mm for equivalent elastic strain, and each of our five samples of R2 varies from under 0.09 up to 0.13. However, deformed strip ends look slightly different between real-life samples and FEM ones. The difference could come from the linear deformation recorded in the tensile tests versus the total deformation in the FEM that considers all six degrees of freedom.

For the R12 dovetail-shaped test sample, the results are shown in Figure 9. The test sample registers around 0.15 mm/mm for equivalent elastic strain, and each of our five samples of R12 vary from under 0.09 up to 0.16. Real-life samples show more severe damage than FEM ones. A possible explanation could be that FEM stops after failure criteria are met, whereas tensile tests continue.

A certain correspondence between real-life samples and FEM results was achieved regarding deformation and failure. The authors acknowledge that further refinement is mandatory in view of the behavior of the material in our experiments, and FEM setup having focused only on female parts.

### 2.5. Using the ANOVA Method to Identify Significant Input Factors

To obtain information on the significance of the factors considered in terms of their influence on the values of the output parameters of the considered process, the ANOVA method was used. Analysis was performed using Minitab software.

Thus, in the case of using the ANOVA method to evaluate the significance of the factors capable of exerting influence on the average force at peak, the results presented synthetically in Table 3 were obtained.

The critical value Fcrit of Fisher’s criterion for a risk coefficient α = 5% was determined as follows. Given the number of degrees of freedom v_1_ = N − k − 1 = 18 − 8 − 1 = 9, and v_2_ = N(n − 1) = 18(8 − 1) = 126, one can evaluate Fcrit = 2.01 from a table that includes Fisher distribution values [26]. Analyzing the values calculated using the Minitab software for *F*-value, it is found that values higher than the critical value (Fcrit = 2.01) were obtained, depending on the values of the *F*-value criterion for overlap *o* (*F*-value = 529.18), material pairs *m* (*F*-value = 111.76), joint shape *j* (*F*-value = 84.65), line width *l* (*F*-value = 42.51), number of walls *no_w_* (*F*-value = 18.01), neck height *n_h_* (*F*-value = 4.76). This means that the six input factors influence the average displacement value at peak *L*.

Regarding the *p*-value, it can be seen that values lower than 0.05 were determined for the following factors: joint shape j (*p*-value = 0.012), materials pairs m (*p*-value = 0.009), line width *l* (*p*-value = 0.023) and overlap o (*p*-value = 0.002). According to current conventions, for *p*-value < 0.05, it can be considered that there is a statistically significant relationship between the input factors for which the condition for *p*-value is met and the output parameter considered (in the present case, the average displacement at peak *L*).

## 3. Results

The experimental results were entered in the last two columns of Table 1. To establish some empirical mathematical models that highlight the influence exerted by the values of the input factors in the investigated process on the values of some output parameters of the same process, specialized software was used for the mathematical processing of experimental results [27]. This software is based on the least squares method. It allows the selection of the most appropriate empirical mathematical model from five such models (first-degree polynomial, second-degree polynomial, power type function, exponential function, and hyperbolic function). The selection of the most appropriate empirical mathematical model to the experimental results is performed using the value of Gauss’s criterion *S_G_*. The *S_G_* value is determined by taking into account the sum of the squares of the differences between the values of the ordinates determined using the proposed empirical model, on the one hand, and the values of the ordinates corresponding to the experimental tests, on the other—for the same values of the abscissas. The smaller the value of Gauss’s criterion, the more appropriate to the experimental results is the empirical mathematical model.

Empirical power function mathematical models have been relatively frequently used in manufacturing processes (for example, to highlight the influence exerted by various factors on the cutting tool life, the size of the cutting forces, the value of some roughness parameters, etc.). Moreover, empirical mathematical power function models can also provide a direct picture of the intensity of influence exerted by a factor on the value of an output parameter by examining the value of the exponent attached to that factor in the power function model. For these reasons, attention was also paid to power functions as empirical mathematical models in the present work. It should be noted that power function-type mathematical models can only be used for a monotonic variation of the output parameter to the values of the input factors. Therefore, it will be assumed that, when pursuing the identification of some power function type mathematical models, and for the various intervals of the values of the input factors, monotonous variations of the two output parameters of the investigated process will then be obtained.

Through the mathematical processing of the experimental results with the help of specialized software, the following empirical mathematical models were arrived at and considered to be the most appropriate for the available experimental results:

For average force at peak:*F* = −54.794 + 87.207*j* − 29.115*j*^2^ + 2.003*m* − 0.501*m*^2^ + 0.0131*n_w_* − 0.000686*n_w_*^2^ − 0.267*n_h_*+ 0.0171*n_h_*^2^ + 0.00602*s* − 0.00121*s*^2^ + 1.996*no_w_* − 0.*265no_w_^2^* − 31.306*l* + 33.259*l*^2^ + 0.306*o* − 0.0261*o*^2^,(1)
for which the value of the Gauss’s criterion is *S_G_* = 0.002434847.

For average displacement at peak:*L* = −258.788 + 443.982*j* − 148.243*j*^2^ − 10.630*m* + 2.625*m*^2^ − 0.169*n_w_* + 0.0239*n_w_*^2^ − 0.425*n_h_*+ 0.0272*n_h_*^2^ + 0.416*s* − 0.0132*s*^2^ + 1.794*no_w_* − 0.303*no_w_*^2^ − 126.527*l* + 138.649*l*^2^ − 0.482*o* + 0.206*o*^2^,(2)
for which the value of Gauss’s criterion is *S_G_* = 0.1335882.

It can be seen that from the point of view of the adequacy of the empirical mathematical models to the experimental results, the most convenient of the five models considered is the polynomial one of the second degree.

If this chosen model considers the determination of mathematical models of power-type functions, it obtains:

For average force et peak:*F* = 0.553*j* − 0.692*m* − 0.0351*n_w_* − 0.105*n_h_* − 0.204*s* − 0.0159*no_w_* − 0.750*l*− 2.556*o*^0^.^143^,(3)
for which the value of Gauss’s criterion is *S_G_* = 0.1720331.

For average displacement at peak:*L* = 7.083*j* − 0.377*m* − 0.204*n_w_* − 0.295*n_h_* − 0.0853*s* − 0.0265*no_w_* − 1.054*l* − 0.0793*o*^0^.^0318^,(4)
for which the value of Gauss’s criterion is *S_G_* = 4.216496.

Using the empirical mathematical models, the diagrams in Figure 10, Figure 11, Figure 12, Figure 13 and Figure 14 were developed. When generating these diagrams, the empirical mathematical models of the second-degree polynomial type (Equations (1) and (2)) were used, estimating that these models better reflect the actual behavior of the material combinations used to make omega and dovetail joints.

## 4. Discussion

The analysis of the empirical mathematical models constituted by Equations (1)–(4) and the graphic representations in Figure 10, Figure 11, Figure 12, Figure 13 and Figure 14 allowed the formulation of the following observations.

It is found that in the mathematical model of the second-degree polynomial corresponding to the output parameter (Equation (1)) of average force *F* at peak, the lowest values of the coefficients attached to the variables *n_w_* and *s* also have the lowest values. This means that the input factors related to these variables (neck width *n_w_* and socket height *s*) practically do not influence the average force *F*. A similar statement can be made about the influence of the two factors (neck width *n_w_* and socket height *s*) on average displacement *L* if the structure of Equation (2) is analyzed. The latter was also highlighted by the results from the ANOVA method.

In the bar diagrams in Figure 10, the values of the average force *F* and average displacement *L* were highlighted for the two types of joints (omega and dovetail) and the three combinations of two materials considered. It is found that those combinations of materials *m* that lead to a higher value of average force *F* are found both in the case of the omega-type joint and in the case of the dovetail-type joint (it is the ABS-PETG combination, corresponding to the symbol *m* = 2). The order of material combinations in terms of average force *F* is the same for both types of joints (omega and dovetail).

It is also observed that omega-type joints lead to higher average forces *F* at peak than those in the case of dovetail-type joints. A possible explanation of this could be based on the existence, in the case of the dovetail joint, of some areas of jointing of certain surfaces at sharp angles, which would lead to the formation of higher stress concentrators and, therefore could reduce the mechanical strength of the joint.

Considering the average displacement *L* at peak, it is found that the lowest displacement values correspond to the same combination of materials that also leads to a maximum value of average force *F* at peak. This is logical because the material that requires the highest value of average force *F* will also have the lowest average displacement *L*.

The existence of maxima or minima for the variations of the *F* and *L* parameters when variations in the values of the significant input factors occur have been highlighted in the diagrams in Figure 11, Figure 12, Figure 13 and Figure 14.

The analysis of the empirical mathematical models constituted by Equations (1) and (2) and the graphic representations in Figure 11, Figure 12, Figure 13 and Figure 14 both highlight the fact that, if we abstract from the type of joint *j* and the pair of materials *m*, the strongest influences on the two output parameters (*F* and *L*) seem to be exerted by the values of the input factors overlap *o*, line width *l,* and the number of walls *w*. These input factors were assigned larger coefficient values from mathematical models (1) and (2). It was thus expected that an increase in the overlap value *o* would lead to higher average breaking forces *F* of the test samples due to the interwoven joining on larger surfaces of the two components of the test samples.

## 5. Conclusions

The study of the information from the specialized literature and industrial practice highlighted the existence of two possibilities for increasing the strength of strip end joints, namely those based on the use of omega-shaped joints and those based on dovetail-type joints. The finite element method was used to simulate the processes in the joint area. As a result of the increase in tensile force, the deterioration process of at least one of the components of the specimen is initiated. In general, the FEM simulation revealed a faster deterioration of the female component of the test samples. The use of three distinct polymeric materials was considered for the realization of the test samples that were later subjected to tensile tests.

Distinct values were proposed and used for some dimensions that characterize omega and dovetail joints, namely the neck width, the neck height, the socket height, and the overlap that characterized the interweaving of some areas of the ends of the joints. The number of walls and the line width were used as input factors in the 3D printing and defined this process. The experimental tests were carried out following the requirements of a Taguchi L18 factorial experiment with 8 independent variables. The average force at the peak and the average displacement at the peak were used as output parameters. In the case of the average displacement, the ANOVA method allowed the identification of the factors that most significantly influence the output parameter values. The experimental results were then mathematically processed with the help of specialized software, and it was found that empirical mathematical models of the second-degree polynomial type are appropriate for the obtained experimental results. The analysis of the empirical mathematical models and the graphic representations developed based on these empirical models highlighted that the main input factors able to exert influence on the values of the output parameters are primarily the type of joint and, secondly, the material components assembled at the joint. In the future, the intention is to expand this theoretical and experimental research by taking into account other polymeric materials and the variation of other test sample dimensions on the values of the two considered output parameters.

## Figures and Tables

**Figure 1 polymers-14-04990-f001:**
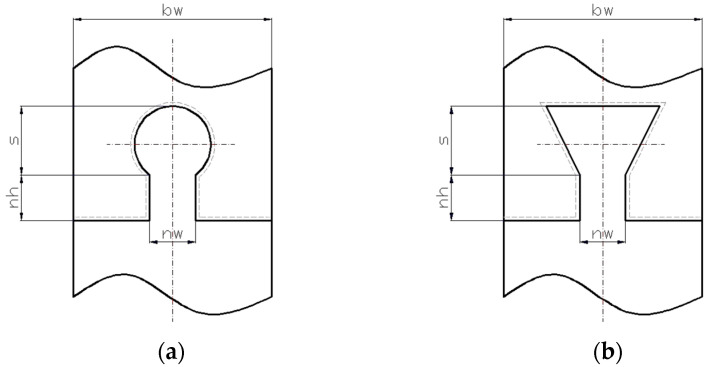
Joints along an outline of the Greek letter omega (**a**) and dovetail (**b**) shape.

**Figure 2 polymers-14-04990-f002:**
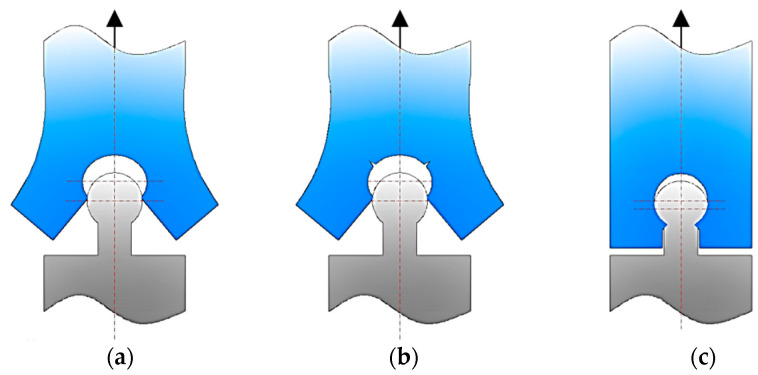
Hypotheses regarding the damage of a joint in the case of an omega-type joint: (**a**) by elastic deformation of some areas of the upper end; (**b**) by deformation until the initiation of cracks in some areas of the upper end; (**c**) by deformation until the initiation of some cracks in the area of the neck on the lower end.

**Figure 3 polymers-14-04990-f003:**
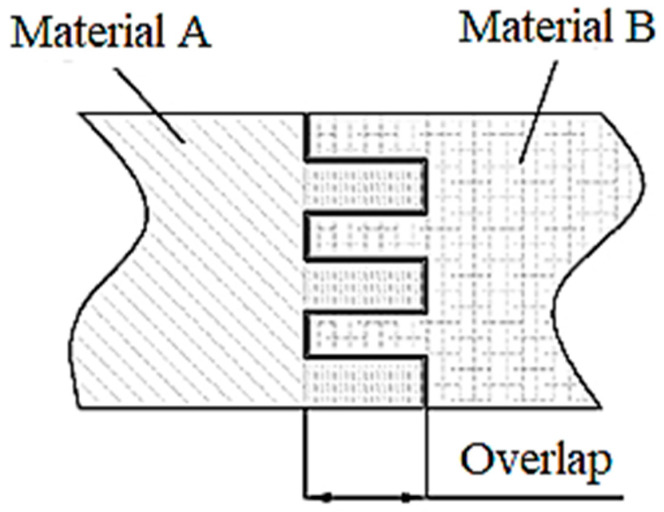
Interlock zone.

**Figure 4 polymers-14-04990-f004:**
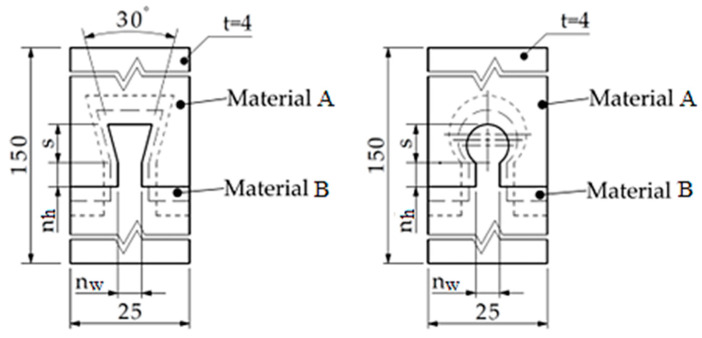
Macroscopic contact interfaces definition on custom samples (design based on type 2 test sample according to ISO 527:3).

**Figure 5 polymers-14-04990-f005:**
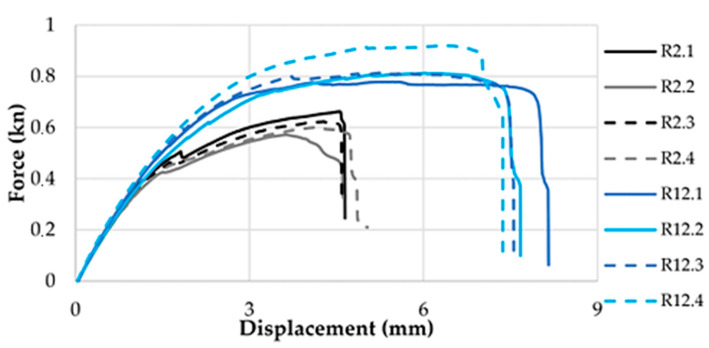
Force-displacement charts for R2 and R12 groups of samples.

**Figure 6 polymers-14-04990-f006:**
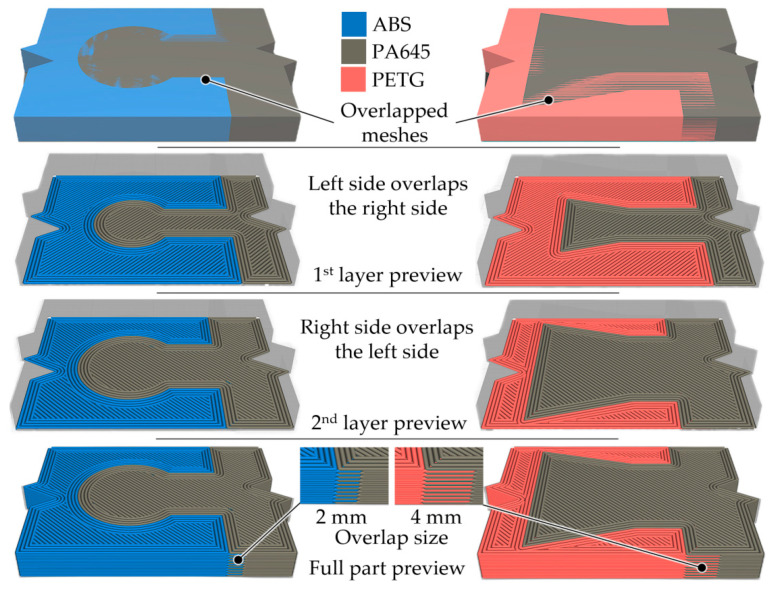
Print preview of the printed samples R2 and R17 experiments from Table 1 using UltimakerCura Arachne Beta 2 engine, with overlap between meshes and alternate mesh removal enabled.

**Figure 7 polymers-14-04990-f007:**
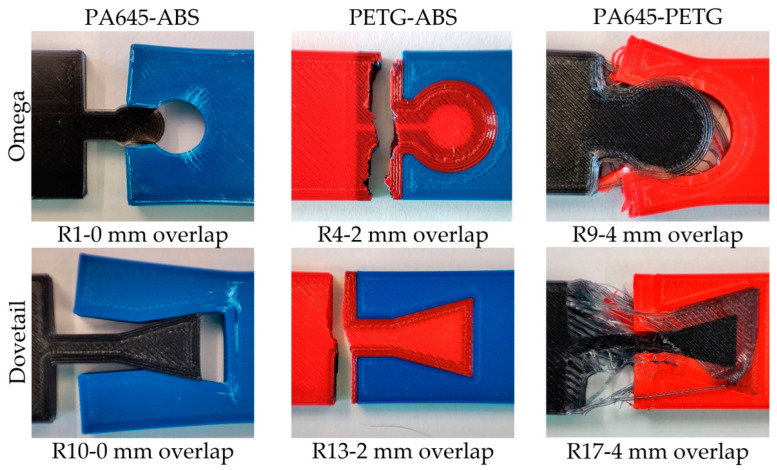
Images of damaged test samples resulting from the tensile test corresponding to experiments R1, R4, R9, R10, R13, and R17.

**Figure 8 polymers-14-04990-f008:**
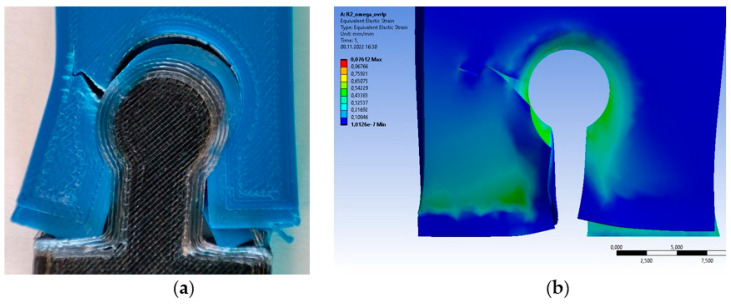
Comparison of results for R2 omega-shaped test sample: (**a**) real-life sample after tensile test; (**b**) FEM result when using the equivalent elastic strain distribution.

**Figure 9 polymers-14-04990-f009:**
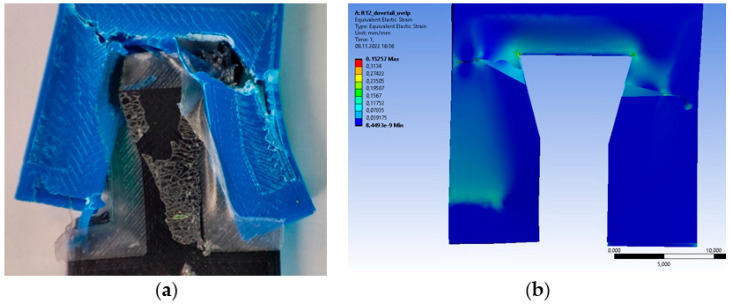
Comparison of results for R12 dovetail-shaped sample: (**a**) real-life sample after tensile test; (**b**) FEM result with equivalent elastic strain distribution.

**Figure 10 polymers-14-04990-f010:**
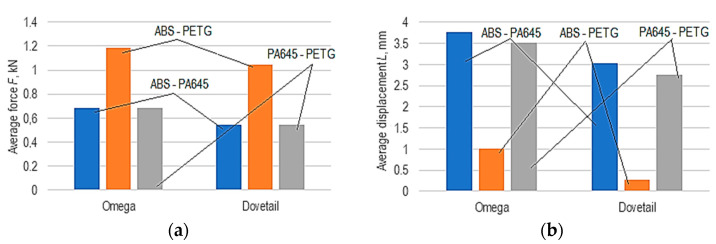
Differences between the corresponding values of: (**a**) average force *F*; (b) average displacement *L* for the three types of joints between each of the two materials (*n_w_* = 8 mm, *n_h_* = 8 mm, *s* = 11 mm, *no_w_* = 4 walls, *l* = 0.45 mm, *o* = 2 mm).

**Figure 11 polymers-14-04990-f011:**
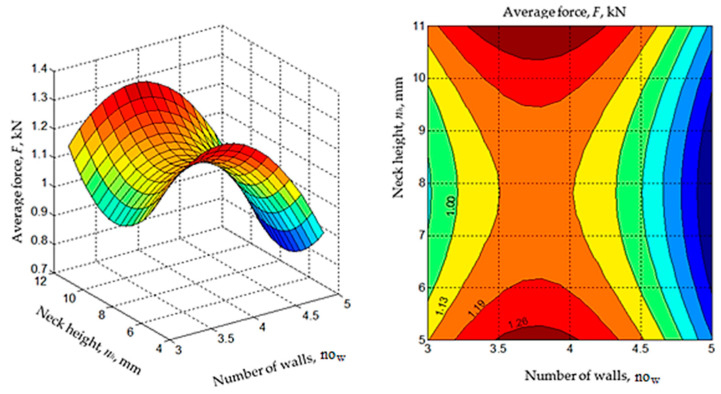
Influence of the neck height *n_h_* and number of walls ***no****_w_* on the average force *F* (*j* = 1 for the omega joint, *m* = 2 for the materials pair ABS-PETG, *n_w_* = 8 mm, *s* = 11 mm, line width *l* = 0.45 mm, overlap *o* = 2 mm; diagrams were made using MATLAB software).

**Figure 12 polymers-14-04990-f012:**
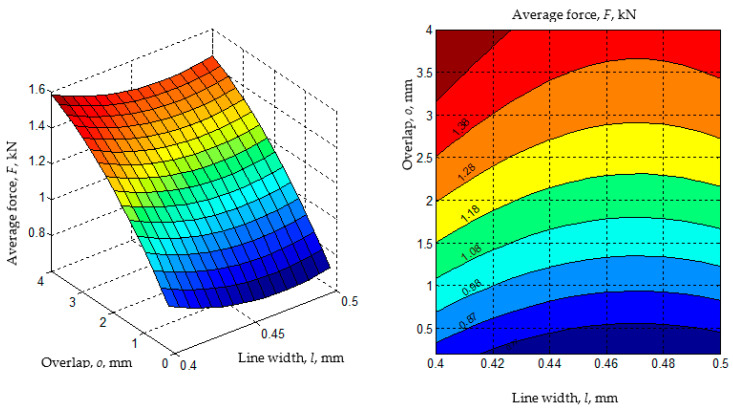
The influence of the line width *l* and overlap *o* on the average displacement *L* (*j* = 1 for the omega joint, *m* = 2 for the materials pair ABS-PETG, *n_w_* = 8 mm, *n_h_* = 8 mm, *s* = 11 mm, *no_w_* = 4 walls; diagrams were made using MATLAB software).

**Figure 13 polymers-14-04990-f013:**
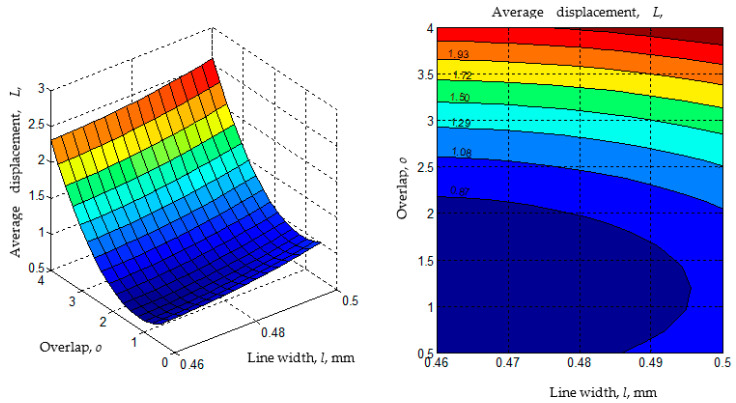
Influence of the overlap *o* and line width *l* on the average displacement *L* (*j* = 1 for the omega joint, *m* = 2 for the materials pair ABS-PETG, *n_w_* = 5 mm, *n_h_* = 5 mm, *s* = 11 mm, *no**_w_* = 4 walls; diagrams were made using MATLAB software).

**Figure 14 polymers-14-04990-f014:**
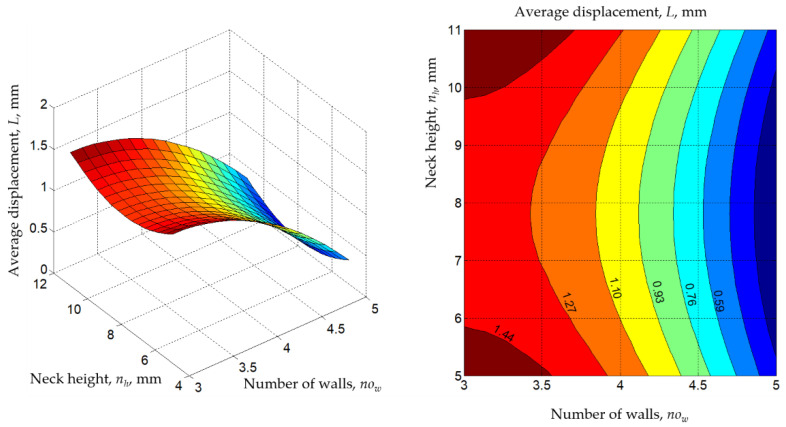
Influence of the number of walls *no_w_* and neck height *n_h_* on the average displacement *L* (*j* = 1 for the omega joint, *m* = 2 for the materials pair ABS-PETG, *n_w_* = 8 mm, *s* = 11 mm, line width *l* = 0.45 mm, overlap *o* = 2 mm; diagrams were made using MATLAB software).

**Table 1 polymers-14-04990-t001:** Experimental conditions and results.

Exp. Run	Input Factors								Output Factors	
	Joint Shape, *j*	Material Pairs, *m*	Neck Width, *n_w_,* mm	Neck Height, *n_h_*, mm	Socket Height, *s,* (mm)	No. of Walls, *no_w_*	Line Width, *l*, (mm)	Overlap, *o*, mm	Avg. Force at Peak, *F*, kN	Avg. Displacement at Peak, *L*, mm
R1	Omega	ABS-PA645	5	5	8	3	0.4	0	0.288	4.330
R2	Omega	ABS-PA645	8	8	11	4	0.45	2	0.613	4.128
R3	Omega	ABS-PA645	11	11	14	5	0.5	4	0.638	6.363
R4	Omega	ABS-PETG	5	5	11	4	0.5	4	1.628	2.253
R5	Omega	ABS-PETG	8	8	14	5	0.4	0	0.373	0.573
R6	Omega	ABS-PETG	11	11	8	3	0.45	2	1.278	1.640
R7	Omega	PA645-PETG	5	8	8	5	0.45	4	0.555	3.095
R8	Omega	PA645-PETG	8	11	11	3	0.5	0	0.148	4.448
R9	Omega	PA645-PETG	11	5	14	4	0.4	2	0.820	5.200
R10	Dovetail	ABS-PA645	5	11	14	4	0.45	0	0.100	2.860
R11	Dovetail	ABS-PA645	8	5	8	5	0.5	2	0.328	1.660
R12	Dovetail	ABS-PA645	11	8	11	3	0.4	4	0.830	5.758
R13	Dovetail	ABS-PETG	5	8	14	3	0.5	2	0.728	0.975
R14	Dovetail	ABS-PETG	8	11	8	4	0.4	4	1.613	2.273
R15	Dovetail	ABS-PETG	11	5	11	5	0.45	0	0.175	0.810
R16	Dovetail	PA645-PETG	5	11	11	5	0.4	2	0.443	2.123
R17	Dovetail	PA645-PETG	8	5	14	3	0.45	4	0.738	5.070
R18	Dovetail	PA645-PETG	11	8	8	4	0.5	0	0.078	3.530

**Table 2 polymers-14-04990-t002:** Constant process parameters and their levels.

Parameter	Value	Parameter	Value
(1) Layer thickness (mm)	0.2	(11) Initial layer speed (mm/s)	15
(2) No. of top/bottom layers	5	(12) Z Hop height (mm)	1.6
(3) Left extruder seam alignment	Back left	(13) Retraction distance (mm)	7
(4) Right extruder seam alignment	Back right	(14) Retraction speed (mm/s)	35
(5) Infill pattern	Gyroid	(15)^1^ Fan speed (%)	10|*15*|30
(6) Infill density (%)	40	(16) Brim	Yes
(7) Printing temperature (°C)	245	(16) Brim width (mm)	3
(8) Bed temperature (°C)	65	(17) Closed environment	Yes
(9) Print speed (mm/s)	30	(18) Alternate mesh removal	Yes

(15)^1^ Normal font is associated with ABS, Italic with PA645, and Bold with PETG.

**Table 3 polymers-14-04990-t003:** Results of analysis of variance for average maximum force.

Source	DF	Seq SS	Adj SS	Adj MS	*F*-Value	*p*-Value
Joint shape, *j*	1	58.07	58.068	58.68	84.65	0.012
Materials pairs, *m*	2	152.32	153.318	76.659	111.76	0.009
Neck width, *n_w_*	2	2.19	2.190	1.095	1.60	0.385
Neck height, *n_h_*	2	6.52	6.524	3.262	4.76	0.174
Socket height, *s*	2	0.32	0.319	0.160	0.23	0.811
Number of walls, *no_w_*	2	24.70	24.701	12.351	18.01	0.053
Line width, *l*	2	58.32	56.324	29.162	42.51	0.023
Overlap, *o*	2	725.99	725.986	362.993	529.18	0.002
Residual error	2	1.37	1.372	0.686		
Total	17	1030.80				

In Table 3, the symbols used have the following meanings: DF—degrees of freedom; Seq SS—Sequential sums of squares; Adj SS—the adjusted sum of squares; Adj MS—adjusted mean squares; *F*-value—value on the *F* distribution; *p*-value—the probability of obtaining the observed results.

## Data Availability

The data presented in this study are available on request from the corresponding author.

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
