# Peer review of "Tensile Behavior of Joints of Strip Ends Made of Polymeric Materials"

_polymers, 2022, doi:10.3390/polym14224990_

Round 1
Reviewer 1 Report
* Title, abstract and keywords are compatible with the content of the article and inform the reader about the study.
* In the introduction, current examples in the literature supporting the study are given.
* It may be better if a single order is followed in the explanations of the figure and figure in the text, first the description in the text and then the figure.
* Table 1 Parentheses used incorrectly in neck width unit.
* It might be a better whole if Figure 4 is given under the heading of Experimental conditions.
* Finite element analysis, which supports experimental analysis, increases the reliability of the study.
* A clearer image can be selected for Figure 11.
* Descriptions of figures and tables are consistent with the text.
* Scientifically successful study has been achieved.
* Scientific, clear and understandable language was used in the study.
Author Response
Authors 'responses to the reviewers' comments
The authors of the reviewed paper wish to express their gratitude for the efforts of the reviewers invested in the analysis of the proposed article and for the useful observations and suggestions to improve the quality of the paper.
REVIEWER 1
Reviewer's comments nos. 1 and 2
* Title, abstract, and keywords are compatible with the article's content and inform the reader about the study.
* In the introduction, current examples in the literature supporting the study are given.
Authors response to the reviewer's comment. The authors of the paper thank the reviewer for the comment.
Reviewer's comment no. 3
* Table 1 Parentheses used incorrectly in neck width unit.
Authors response to the reviewer's comment. The reviewer is right. Removed parenthesis.
Reviewer's comment no. 4. * It might be a better whole if Figure 4 is given under the heading of Experimental conditions.
Authors response to the reviewer's comment. The authors considered that the reviewer was correct and made the proposed change, but in the final stage, the editors will be able to place the figures as they see fit.
Reviewer's comment no. 5. Finite element analysis, which supports experimental analysis, increases the reliability of the study
Authors response to the reviewer's comment. The authors of the paper thank the reviewer for their comments.
Reviewer's comment no. 6. * A clearer image can be selected for Figure 11.
Authors response to the reviewer's comment. The authors considered that the reviewer was correct. Replaced figures 11-14 with improved versions.
Reviewer's comment no. 7. * Descriptions of figures and tables are consistent with the text.
Authors response to the reviewer's comment. The authors of the paper thank the reviewer for the comment.
Reviewer's comment no. 8. * Scientifically successful study has been achieved.
Authors response to the reviewer's comment. The authors of the paper thank the reviewer for the comment.
Reviewer's comment no. 7. * Scientific, clear and understandable language was used in the study.
Authors response to the reviewer's comment. The authors of the paper thank the reviewer for the comment.

Reviewer 2 Report
The authors use 3D printed method to create strip joints of different geometries, then evaluated their strength via tensile experiments. The experiment is sound, while the finite element analysis and data processing require more clarifications:
1. More explanation on FEA model set up is needed. How does the author define the interwoven interface between two parts? If not interwoven, what is the contact property at the joint?
2. What is the crack initiation criterion in FEA? From the strain contour, it is unclear where is the spot with highest strain. It is surprising that it is not located at the crack tip.
3. Does the model start from a symmetric geometry with uniform material property? If so, what causes the asymmetric deformed configuration?
4. For the empirical fitting (eq. 1 to 4), how many data points are used? How did the author check the range of validation of fitting?
5. Considering that these experiment is performed on 3D printed parts, which is anisotropic due to layer-to-layer printing, how is the understanding generated by this study transferable to application?
Author Response
Authors 'responses to the reviewers' comments
The authors of the reviewed paper wish to express their gratitude for the efforts of the reviewers invested in the analysis of the proposed article and for the useful observations and suggestions to improve the quality of the paper.
REVIEWER 2
Reviewer's comment no. 1. More explanation on FEA model set up is needed. How does the author define the interwoven interface between two parts? If not interwoven, what is the contact property at the joint?
Authors response to the reviewer's comment. There are no interwoven interfaces between parts. Strip ends of 0.2mm in height were designed to overlap on top of each other with corresponding gaps between the female and male parts. As a result, both experimental tests and FEM analyses considered only the friction between strip ends with a friction coefficient of 0.2 at dynamic runs according to [https://www.tribology-abc.com/abc/cof.htm] and [https://www.engineeringtoolbox.com/friction-coefficients-d_778.html]. But as the Ansys Explicit Dynamic failed due to poor computing power, the new Static Structural did not require a contact joint because only the female part was put to the test. We have therefore imposed fixed supports and force loads to each of the ten strip ends in the case of both omega and dovetail samples. By applying a single force on the top surface of the female sample, we did not achieve satisfying results in the case of crack growth as it manifests only on a single end strip as the others are deformed. To be able to get results both for crack growth and deformation of all end strips in the same analysis, we decided to treat each strip end individually within the same timeframe
Reviewer's comment no. 2. What is the crack initiation criterion in FEA? From the strain contour, it is unclear where is the spot with highest strain. It is surprising that it is not located at the crack tip.
Authors response to the reviewer's comment. Within Ansys Static Structural, we have used the Fracture tool with the SMART Crack Growth condition. The aim was to see if the crack is propagating the same as it did in experimental tests. We have measured the R2 and R12 samples after tensile tests, for heights at which cracks initiate. Therefore, we have designed small V-shaped 3D cracks for the pre-mesh processes. All cracks receive their own Coordinate Systems, which respect the need for the crack to propagate on the desired direction as it does in real-life, in the sense that the X axis points to the initial crack growth direction as the Y is almost normal to the top face and -Y sits almost normal to the bottom one. Each crack’s own Coordinate System was considered inside Pre-Meshed Crack tool with six solution contours. A SMART Crack Growth condition was considered based on Paris law with the option set to Fatigue. The methodology for crack growth was set to Cycle by Cycle with increments of 100000 cycles. We have not stopped at Maximum Crack Extension. Fracture Controls in Analyses Settings were on with Stress Intensity Factor enabled.
The areas which contain the highest strains sit on the inside faces of omega and dovetail shapes as they deform by being pulled upwards. This is due to the fact that strip ends start to deform before crack propagation finishes because the analysis stops after SMART re-mesher fails to produce the new mesh with good quality. If a single strip were to be considered, the highest strain would be at the tip of the crack, as presumed
Reviewer's comment no. 3. Does the model start from a symmetric geometry with uniform material property? If so, what causes the asymmetric deformed configuration?
Authors response to the reviewer's comment. ABS was chosen from the Ansys Granta library. PETG was developed based on PET Plastic. Both materials consider nonlinear behaviors. This is equivalent to the anisotropy property of materials which manifests as directionally dependent as opposed to isotropy which gives homogeneity in all directions. That is why we have introduced a separate Coordinate System for each crack to be able to follow the propagation along a certain axis. We have used symmetric 3D models in all our analyses. But because we aimed to achieve results as close to the ones obtained from experimental tensile tests in terms of crack propagation, we have treated each strip end individually. Thus, the strip ends that are not cracked have received force loads inclined in the opposite direction of the crack’s expected direction of propagation. This was achieved by using the crack’s designated Coordinate System for all crack-free strip ends. That has resulted in an asymmetric response due to the addition of a Nonlinear Adaptive Region tool from Ansys. The criterion was set to Mesh with the option for Skewness and Jacobian ratio with default values. Otherwise, the model reacts symmetrically, which is inconsistent with real-life results for our considered samples as the crack’s shape and length are no longer adequate.
Reviewer's comment no. 4. For the empirical fitting (eq. 1 to 4), how many data points are used? How did the author check the range of validation of fitting?
Authors response to the reviewer's comment. In table 1, the results presented in the last two columns (average forces and average displacements) are averages obtained from performing 4 tests. In total, a number of n=4x18=72 samples were used for the experimental tests. A proper validation of the empirical models was not performed. The authors of the paper thank the reviewer for the formulated observation/suggestion and, in the next period, will consider the realization of additional experimental tests, the results of which will allow the validation of the proposed empirical models.
Reviewer's comment no. 5. Considering that these experiment is performed on 3D printed parts, which is anisotropic due to layer-to-layer printing, how is the understanding generated by this study transferable to application?
Authors response to the reviewer's comment. Even if FFF 3D printed parts have a high anisotropy compared to other printing, some limitations could be diminished using certain methods for designing for manufacturing and load directions. Some researchers, such as Ribeiro and Sousa Carneiro (reference no. [5] in the list of references), and Kuipers et al. (reference no. [9]), reported the benefits of a macroscopic interface when printing low-compatibility materials [5, 9]. Other studies investigated the influence of the interface’s shape on mechanical properties [6, 19]. They showed that it positively influences interface strength by generating zones of horizontal adhesion in addition to vertical ones.
On the other hand, an overlap or a superposition between the mating materials also improves strength [8]. As presented in Table 1, the experimental configurations without overlap showed a small load capacity (i.e., R1 and R10 for ABS-PA645, R5 and R15 for ABS-PETG, and R6 and R18 for PA645- PETG) compared to the overlapped configurations. The high overlap between sample bodies (i.e., 4mm) is printed as a woven (stacked) structure at the level of the interface (see Figure 6). In this way, in addition to the macroscopic interlocking, the load capacity is improved by the friction force generated by the stacked layers of material.
As a general remark, the overlap between parts bodies improves the failure mode (see figure 7) and the load capacity (see table 1) regardless of the contact shape dimensions. This shows that higher superpositions (e.g., min 2 mm) could be used even in regular contact interfaces to increase strength and provide ideas for new applications.
AUTHORS' RESPONSES TO ASPECTS OF THE PAPER THAT NEEDED IMPROVEMENT
Is the research design appropriate? (can be improved)
Authors response to the reviewer's comment. The paper's authors have considered the reviewer's comment and made some changes/completions in the article.
Are the methods adequately described? (must be improved)
Authors response to the reviewer's comment. The paper's authors considered the reviewer's comment and included additional information regarding the use of FEM.
Are the results clearly presented? (can be improved)
Authors response to the reviewer's comment. The paper's authors have included improved versions of Figures 11-14.
Are the conclusions supported by the results? (can be improved)
Authors response to the reviewer's comment. The authors considered the reviewer's comment and included additional conclusions.

Reviewer 3 Report
This paper mainly use the finite element method(FEM) to research the influence of the shape and dimensions of some joints, along with the values of some input factors in the 3D printing process, on the tensile strength of some samples made of pairs of distinct polymers. The content is substantial and the paper has high practicability. But, there are some small mistakes need to revise.
1. There are too many keywords, and just need list the most important ones. In addition, keywords are not simplified enough.
2. In the abstract, the author make a list of examples. However, it is not necessary to put one example in a paragraph, you can put them together.
3. In the abstract, the author make a list of examples. And what’s the deficiencies in these cases, and what are we doing about them? In short, what is the innovation of this article?
4. In line 144-145, the author said “However, research showed that blends of ABS and PETG were more successfully blended with beer properties than the base materials [11].” The author need to explain what is the “beer properties”?
5. Many words and sentences are green and the author needs to change them to black. Many sentences have been marked with strikeouts, but still appear in the article and need to be addressed.
6. Usually, researchers use the control variable method to study the impact of a factor on the final performance. But in Table 1, many factors are like random inputs. In other words, just from the Table 1, we can't see the influence of input factors on the average force and displacements at all, so can you explain it.
7. In line 242, the phrase “ Uniaxial Tension Test Data” Doesn't need to capitalize the first letter.
8. In line 169, the author said “Four replicates were printed for each run covered in the experimental array.”, however in line 251 the author said “and each of our five samples of R2 vary from under 0.09 up to 0.13.”, so why is the number of tests inconsistent?
9. In Figure 10, does author have test the average force and average displacement of three types of joints with nw=8 mm, nh=8 mm, s=11 mm, w=4 walls, l=0.45 mm, o=2 mm? Because most of the data ( average force and average displacement) can’t found in Table 1.
10. The title of Figure 12 and Figure 14 are wrong, and they don't agree with the picture.
11. In Figure 12, in the title the author use “w” represent the line width. However, in the picture, the author use “l” represent the line width.
12. In line 408, the word “of” need to delete.
13. The conclusions are too jumbled and need to be condensed.
Author Response
Authors responses to the reviewer comments
The authors express their gratitude for the reviewer's comments and suggestions for improving the content of the paper.
- Reviewer comments. There are too many keywords, and just need list the most important ones. In addition, keywords are not simplified enough.
Authors response. The reviewer was found to be correct. The requested changes have been made.
- Reviewer comments.In the abstract, the author make a list of examples. However, it is not necessary to put one example in a paragraph, you can put them together.
Authors response. The reviewer is right. The abstract was written in a single paragraph.
- Reviewer comments. In the abstract, the author make a list of examples. And what’s the deficiencies in these cases, and what are we doing about them? In short, what is the innovation of this article?
Authors response. The authors believe that the reviewer is right. The authors of the article considered that there are fewer explanations regarding the behavior of the analyzed joints under tension loads and included an additional explanation in the Abstract (”From the point of view of industrial practice, it is of interest to study the behavior of these joints under stretching requests.”).
- Reviewer comments. In line 144-145, the author said “However, research showed that blends of ABS and PETG were more successfully blended withbeer properties than the base materials [11].” The author need to explain what is the “beer properties”?
Authors response. The reviewer is right. The quoted text contains an editing error ("beer", instead of "better"). The sentence has been modified to express the authors' point more clearly (”However, research has shown that blends of ABS and PETG have better properties than the base materials [11].”)
- Reviewer comments. Many words and sentences are green and the author needs to change them to black. Many sentences have been marked with strikeouts, but still appear in the article and need to be addressed.
Authors response. The authors have used the color green to highlight places in the article where changes have been made based on reviewers' comments. Texts written using green color will be transformed into texts written in black color in the final stage of article editing and any strikeout text will be removed.
- Reviewer comments. Usually, researchers use the control variable method to study the impact of a factor on the final performance. But in Table 1, many factors are like random inputs. In other words, just from the Table 1, we can't see the influence of input factors on the average force and displacements at all, so can you explain it.
Authors response. The way to enter the values of the variables in Table 1 corresponds to the so-called Taguchi L18 array. By using a Taguchi L18 array, it becomes possible to significantly reduce the number of experimental tests needed to be performed to arrive at an empirical mathematical model, including in the case of a number of 8 input factors (8 independent variables). Only by examining the information in Table 1, no observations can be made regarding the influence of some input factors on the values of the output parameters of the studied process. For this reason, it was necessary to mathematically process the information in the table and determine, as such, an empirical mathematical model. Examining the empirical mathematical model and the graphical representations developed by taking into account the empirical mathematical model allows the formulation of some observations regarding the influence of the variation of the input factors on the values of the output parameters.
- Reviewer comments. In line 242, the phrase “ Uniaxial Tension Test Data” Doesn't need to capitalize the first letter.
Authors response. Appreciating that the reviewer is right, lowercase letters have been used in the case of the mentioned concept.
- Reviewer comments. In line 169, the author said “Four replicateswere printed for each run covered in the experimental array.”, however in line 251 the author said “and each of our five samples of R2 vary from under 0.09 up to 0.13.”, so why is the number of tests inconsistent?
Authors response. Considering that the reviewer is right, we have replaced “Four replicates were printed for each run covered in the experimental array.” with “Each run covered in the experimental array was printed in four or five replicates.” and modified for R2 the number of samples to five.
- Reviewer comments. In Figure 10, does author have test the average force and average displacement of three types of joints with nw=8 mm, nh=8 mm, s=11 mm, w=4 walls, l=0.45 mm, o=2 mm? Because most of the data ( average force and average displacement) can’t found in Table 1.
Authors response. The authors intended to show that they determined the average values of force at peak and displacement at peak, and these average values (for F and L) were entered in Table 1. When calculating the average values (entered in the last two columns of Table 1 ), the values obtained through 4 experimental tests were used, corresponding to each of the 18 sets of values of the input factors in the investigated process.
- Reviewer comments. The title of Figure 12 and Figure 14 are wrong, and they don't agree with the picture.
Authors response. The reviewer is correct. We have modified the titles and legends.
- Reviewer comments. In Figure 12, in the title the author use “w” represent the line width. However, in the picture, the author use “l” to represent the line width.
Authors response. The reviewer is right: ”w” has been replaced by ”l” to refer to the line width and now to refer to the number of walls.
- Reviewer comments. In line 408, the word “of” need to delete.
Authors response. The reviewer is right. The word "of" was deleted
- Reviewer comments. The conclusions are too jumbled and need to be condensed.
Authors response. The reviewer was found to be correct. The number of sentences in the conclusions chapter was reduced.

Round 2
Reviewer 2 Report
After reading the author's answers, I have major concern on the methodology used and the soundness of the results as follow:
1. The reviewer mentioned that the crack growth is modeled using the Paris law. This is wrong because Paris law is used to model fatigue crack while the authors are modeling an uniaxial tension process. These two loadings are completely different and the mechanism of crack propagation is also different. The authors should not use Paris theory for their crack growth modeling.
2. When explaining why the crack tip does not have highest stress, the authors wrote "This is due to the fact that strip ends start to deform before crack propagation finishes because the analysis stops after SMART re-mesher fails to produce the new mesh with good quality.". To the reviewer's understanding, this means the simulation crashed due to remesh failure. If that's the case, a non-converged simulation cannot be trustworthy and work needs to be done to fix it. After that mesh convergence is still needed to ensure the result is stable, which I don't think the authors did because the simulation crashed.
3. When answering the question regarding where anisotropy is induced in simulation, the authors answered "Both materials consider nonlinear behaviors. This is equivalent to the anisotropy property of materials which manifests as directionally dependent as opposed to isotropy which gives homogeneity in all directions." This answer is concerning as the authors, first, did not address the question on the cause of anisotropy; second, the author confused the definition of nonlinearity and anisotropy. For instance, neo-hookean material model is nonlinear, isotropic model; while many models on liquid crystal elastomers are nonlinear, anisotropic. The author should just point out what is the cause of anisotropy and what material models were used.
3. The authors mentioned that a total number of 72 datapoints were used to fit eq. 1-4, each of which are highly nonlinear equations with ~8 degrees of freedoms. Given the small size of data space, I am not convinced by the accuracy of this fit. It is very likely that these equations only interpolate the experimental datapoints, but because inaccurate when predicting a new sample condition. The authors need to either significantly increase the number of datapoints, and then perform validation with testings on several new samples. Otherwise, I suggest delete the section on "mathematical model".
Author Response
Authors 'responses to the reviewers' comments
The authors would like to thank the reviewer for his comments and suggestions.
Reviewer's comment no. 1. 1. The reviewer mentioned that the crack growth is modeled using the Paris law. This is wrong because Paris law is used to model fatigue crack while the authors are modeling an uniaxial tension process. These two loadings are completely different and the mechanism of crack propagation is also different. The authors should not use Paris theory for their crack growth modeling.
Authors response to the reviewer's comment. The reviewer is right. A new setup has been used where we have kept the Fracture Mechanics Approach. We have still used a pre-meshed crack. For our analysis, we have chosen the SMART arbitrary path approach for the crack growth, but instead of Fatigue Crack Growth, we have now used Static Crack Growth with Failure Criteria Option set to Stress Intensity Factor with a critical rate of around 100 MPa. Also, in the Engineering Data section, we have deleted Paris Law's dependent parameters. Instead, we have introduced the Uniaxial Tension Test Data from the Hyperelastic Experimental Data section. In a tabular form, we have used values given by the Instron 4411 equipment for experimental tensile tests for stress and strain. Since we have no data for lateral strain, we kept that option to No. We feel lateral strain has a more significant impact, especially on hyperelastic materials. We now have obtained fully converged analyzes for both shapes. New images are proposed inside the manuscript with added explanations. Since strain-related data was used, we felt it more appropriate to use Equivalent Elastic Strain distribution images instead of Total Deformation ones.
Reviewer's comment no. 2. When explaining why the crack tip does not have highest stress, the authors wrote "This is due to the fact that strip ends start to deform before crack propagation finishes because the analysis stops after SMART re-mesher fails to produce the new mesh with good quality.". To the reviewer's understanding, this means the simulation crashed due to remesh failure. If that's the case, a non-converged simulation cannot be trustworthy and work needs to be done to fix it. After that mesh convergence is still needed to ensure the result is stable, which I don't think the authors did because the simulation crashed.
Authors response to the reviewer's comment. The reviewer is right. We have re-run analyses with different setups. However, our FEM analysis goal consists of highlighting a possible distribution of stress/strain as the tensile process unfolds. The results are intermediate and should not be considered inside new research without further refinement. We have obtained a correspondence for strip ends between FEM and the experimental tests regarding failure and deformation. But, since only part of the process is being analyzed, we acknowledge that the results are intermediate. A special mention has been introduced in the manuscript's text regarding this issue.
Reviewer's comment no. 3. When answering the question regarding where anisotropy is induced in simulation, the authors answered "Both materials consider nonlinear behaviors. This is equivalent to the anisotropy property of materials which manifests as directionally dependent as opposed to isotropy which gives homogeneity in all directions." This answer is concerning as the authors, first, did not address the question on the cause of anisotropy; second, the author confused the definition of nonlinearity and anisotropy. For instance, neo-hookean material model is nonlinear, isotropic model; while many models on liquid crystal elastomers are nonlinear, anisotropic. The author should just point out what is the cause of anisotropy and what material models were used.
Authors response to the reviewer's comment. The reviewer is right. New setups are based on the same material from PET plastic retrieved from Ansys’s Granta library with added Uniaxial Tension Test Data. Because insufficient data was available for the Anisotropic Elasticity property, we used the default Isotropic Elasticity property derived from Young’s modulus and Poisson’s ratio for our analyses. A special mention was introduced in the manuscript regarding this issue. A possible reason for the anisotropic-like behavior may be due to the Nonlinear Adaptivity Criteria set to Mesh Quality inside the Adaptive Region tool.
Reviewer's comment no. 4. The authors mentioned that a total number of 72 datapoints were used to fit eq. 1-4, each of which are highly nonlinear equations with ~8 degrees of freedoms. Given the small size of data space, I am not convinced by the accuracy of this fit. It is very likely that these equations only interpolate the experimental datapoints, but because inaccurate when predicting a new sample condition. The authors need to either significantly increase the number of datapoints, and then perform validation with testings on several new samples. Otherwise, I suggest delete the section on "mathematical model".
Authors response to the reviewer's comment. In the field of design of experiments, many years ago, Professor Ronald Fisher proposed the identification of solutions that lead to the reduction of the number of experimental tests necessary to establish an empirical mathematical model capable of highlighting the influence of some input factors in the investigated process on the values of some output parameters. When only the least-squares method is applied, it has been found that a large number of experiments are required to arrive at a sufficiently accurate empirical mathematical model. A significant reduction in the number of experimental trials has become possible by using the so-called design of experiments (design of experiments, https://en.wikipedia.org/wiki/Ronald_Fisher), and several categories of experimental plans have been proposed. In one of the many works that also addresses the issue of factorial experiments (Wayne W. Daniel, Chad L. Cross, Biostatistics: A Foundation for Analysis in the Health Sciences, John Wiley & Sons, 2018, ISBN: 978-1-119-49657- 1), in chapter 8.5 (”The Factorial Experiment”), the authors show that "The advantages of the factorial experiment include the following: 1. The interaction of the factors may be studied. 2. There is a saving of time and effort.".
Information on using the L18 Taguchi factorial experiment can be found in many works on the Internet. A Google search using the terms "Taguchi" AND "L18" reveals over 64,000 web pages that address aspects of the Taguchi L18 array. Just a few examples of this are the following:
- Fraley Stephanie, Zalewski John, Oom Mike, & Terrien Ben, 14.1: Design of experiments via Taguchi methods - orthogonal arrays, 2022, availabe at https://eng.libretexts.org/Bookshelves/Industrial_and_Systems_Engineering/Book%3A_Chemical_Process_Dynamics_and_Controls_(Woolf)/14%3A_Design_of_Experiments/14.01%3A_Design_of_Experiments_via_Taguchi_Methods_-_Orthogonal_Arrays, accessed: 29.09.2022;
- Orthogonal Arrays (Taguchi Designs), 2004, available at: https://www.york.ac.uk/depts/maths/tables/orthogonal.htm, accessed: 29.09.2022;
- N. Kuramochi and Y. Oomuro, Robust design with direct product of the L18 orthogonal arrays, 2008 International Symposium on Semiconductor Manufacturing (ISSM), 2008, pp. 191-194.
- Raghu N. Kacker, Eric S. Lagergren, and James J. Filliben, Taguchi’s orthogonal arrays are classical designs of experiments. J Res Natl Inst Stand Technol., 96(5), 577–591, 1991, etc.
In our paper, a factorial experiment of type L18 was used, with 8 independent variables (one independent variable at two levels of variation and 7 independent variables at three levels of variation). Since there is a relatively large number of experimental points (18 points), according to the principles of design of experiment, it would have been possible for each experiment to be carried out only once. To increase the accuracy of the empirical model, the authors of the analyzed paper resorted, however, to repeating each experiment 4 times, using in calculations the average of the values obtained through the 4 experimental attempts. In this way, 72 experimental trials were reached, although only 18 experimental tests would have been sufficient.
For the reasons stated above, we cannot agree with the reviewer's suggestion to remove the section on empirical mathematical modeling (“Otherwise, I suggest delete the section on "mathematical model"), because our experimental research was carried out in accordance with principles accepted by researchers in the field.
The reviewer is right when he requests the realization of some experimental tests to validate the empirical mathematical models (most of the works published so far regarding the Taguchi array L18 do not mention the realization of additional experimental validation tests), but, in the available time, it was not possible to carry out additional experimental tests. It follows that, in the next period, we will pay attention to this aspect.
